# Robust COVID-19 Vaccine Responses Despite Filarial Co-Infection: Insights from a Lymphatic Filariasis Cohort in Ghana

**DOI:** 10.3390/vaccines13030312

**Published:** 2025-03-13

**Authors:** Julia Meyer, Jennifer Nadal, Linda Batsa Debrah, Alexander Yaw Debrah, Jubin Osei-Mensah, Derrick Adu Mensah, Patricia Jebett Korir, Janina M. Kuehlwein, Ute Klarmann-Schulz, Achim Hoerauf, Tomabu Adjobimey

**Affiliations:** 1Institute of Medical Microbiology, Immunology and Parasitology (IMMIP), University Hospital Bonn, 53127 Bonn, Germany; 2Institute for Medical Biometry, Informatics, and Epidemiology (IMBIE), University Hospital Bonn, 53127 Bonn, Germany; 3Department of Clinical Microbiology, School of Medicine and Dentistry, Kwame Nkrumah University of Science and Technology (KNUST), Kumasi 00233, Ghana; 4Kumasi Centre for Collaborative Research in Tropical Medicine (KCCR), Kwame Nkrumah University of Science and Technology (KNUST), Kumasi 00233, Ghana; 5Faculty of Allied Health Sciences, Kwame Nkrumah University of Science and Technology (KNUST), Kumasi 00233, Ghana; 6Department of Pathobiology, School of Veterinary Medicine, Kwame Nkrumah University of Science and Technology (KNUST), Kumasi 00233, Ghana; 7Department of Medical Laboratory Technology, Royal Ann College of Health, Kumasi 00233, Ghana; 8Bonn-Cologne Site, German Center for Infectious Disease Research (DZIF), 53113 Bonn, Germany; 9Laboratoire de Biologie Intégrative Pour l’Innovation Thérapeutique (BioInov), Faculté des Sciences et Techniques (FAST), Université d’Abomey Calavi, Abomey Calavi BP 526, Benin

**Keywords:** COVID-19 vaccination, SARS-CoV-2 seroprevalence, lymphatic filariasis, immune response, co-infection, *Ascaris lumbricoides*, IgA/IgG antibodies, Ghana

## Abstract

Background/Objectives: Although the COVID-19 pandemic has largely concluded, the varied trajectories it has followed in different regions of the world remain incompletely understood. Intensive research is needed to fully grasp its course and the implications for future global health challenges. Notably, the milder trajectory of the COVID-19 pandemic in Sub-Saharan Africa has defied initial predictions. An emerging body of evidence suggests that, in addition to the continent’s younger average age and the lower prevalence of relevant comorbidities, co-infections with helminths may have also impressively shaped the pandemic’s milder trajectory in the region. Indeed, helminths are renowned for their ability to modulate human immune responses, which, while potentially beneficial in limiting excessive inflammation, could also diminish vaccine efficacy and impede viral clearance. This study investigated different aspects of the intricate interactions between COVID-19 and Lymphatic Filariasis (LF), a helminth infection caused by parasitic worms such as *Wuchereria bancrofti*, *Brugia malayi*, and *Brugia timori* and endemic to various regions in Sub-Saharan Africa and the tropics. Methods: For this purpose, samples of a larger and ongoing clinical trial (ethical approval codes: CHRPE/AP/525/17 and 325/21; trial registration number ISRCTN14042737) were collected from 222 individuals from endemic areas of Ghana, along with comprehensive clinical and demographic data. The samples include LF patients (*n* = 222) grouped according to their Lymphoedema (LE) stages, as well as COVID-19 vaccinated (*n* = 81) and non-vaccinated individuals (*n* = 141). All vaccinated participants received the COVID-19 vaccine ChAdOx1-S (also known as Vaxzevria) developed by the University of Oxford and AstraZenca. The expressions of SARS-CoV-2 and filarial-specific antibodies (IgG, IgA) were accessed using ELISA, while Luminex-based immunoassays were employed to measure the expression of SARS-CoV-2 variant-specific neutralizing antibodies. The interplay between vaccine responses and demographic factors was analyzed using group comparisons with the Kruskal-Wallis or Mann-Whitney U tests. Results: The results indicate that a remarkable portion of unvaccinated individuals (56% IgA seropositive, 39% IgG seropositive) developed antibodies against SARS-CoV-2 despite no confirmed infection. Notably, the study identified a robust antibody response to COVID-19 vaccination, which was independent of the degree of LF pathology or parasitic status. An important observation was the reduced SARS-CoV-2 antibody response in individuals seropositive for *Ascaris lumbricoides* (*p* = 0.0264), highlighting an interaction between roundworm infection and COVID-19. Conclusions: The study concludes that the ChAdOx1-S COVID-19 vaccine (AstraZeneca) triggers a strong immune response in LF patients; however, filarial and/or soil-transmitted helminth seropositivity might influence the COVID-19 infection-induced response. These findings emphasize the complexity of infectious disease dynamics in co-infected populations and the need to decipher parasite-induced immunomodulatory mechanisms on COVID-19 vaccination.

## 1. Introduction

The COVID-19 pandemic, caused by the coronavirus SARS-CoV-2, has been a defining global health crisis of the early 21st century. Its impact has been profound and widespread, affecting every aspect of human life [1,2,3,4,5,6]. As of 11 January 2025, 777,126,421 cases were reported to the WHO, along with 7,079,925 reported COVID-19 deaths globally [7]. Most people infected with the virus display mild to moderate symptoms such as fever, cough, shortness of breath, muscle aches, headache, loss of taste and smell, sore throat, congestion, or runny nose [8]. However, research has indicated that individuals with advanced age, male gender, and comorbidities or certain pre-existing conditions, such as cardiovascular disease, diabetes, chronic respiratory disease, or cancer, may have an elevated risk of developing severe responses to COVID-19 due to underlying chronic inflammation or immune system imbalances that might lead to the requisition of medical intervention or even death [8,9,10,11]. Approximately 6 to 10% of COVID-19 patients fall into this category and require intensive care, with a significant portion needing mechanical ventilation due to severe respiratory distress [12,13]. The virus enters host cells through the ACE2 receptor, triggering an innate immune response characterized by the production of type I interferons and pro-inflammatory cytokines [14,15]. If the innate response is insufficient, the adaptive immune system is activated, involving T cells and B cells, which produce antibodies to neutralize the virus [16,17,18,19]. In severe cases, a dysregulated immune response can occur, often resulting in a “cytokine storm”, characterized by the release of large amounts of pro-inflammatory cytokines and chemokines, such as interleukin-6 (IL-6), tumor necrosis factor-alpha (TNF-α), and interleukin-1beta (IL-1β), which can cause widespread tissue damage leading to acute respiratory distress syndrome (ARDS) and multi-organ failure [14,16]. The cytokine storm is not only a marker of severe disease but also a critical determinant of mortality in COVID-19 patients. In the fight against the pandemic, a critical tool in reducing the severity and mortality has been the development of vaccines. Various platforms have been employed, including mRNA, viral vector-based DNA, and attenuated virus vaccines. Each of these platforms offers unique attributes and immunogenic properties, as well as differing potentials for adverse reactions. mRNA vaccines, such as those developed by Pfizer-BioNTech and Moderna, are notable for their rapid design and production capabilities, which have proven especially useful during the pandemic [20]. Viral vector vaccines, such as those from AstraZeneca and Johnson & Johnson, utilize modified viruses to deliver genetic material that induces an immune response [21]. Attenuated virus vaccines use a weakened form of the virus to stimulate immunity without causing disease [21]. These diverse approaches have collectively contributed to the global effort to control COVID-19 through vaccination [21]. Globally, as recorded by Our World Data on 11 January 2025, over 13.72 billion doses of COVID-19 vaccines have been administered, with approximately 1.43 billion doses distributed in the European region and 874 million doses in the African region. Notably, 70.7% of the world’s population has received at least one dose of a COVID-19 vaccine. However, this coverage is significantly lower in low-income countries, where only 32.8% of people have received at least one dose [22]. Despite having a low vaccine coverage, African countries have demonstrated a relatively low number of severe cases of COVID-19 [7]. This discrepancy raises important questions about the factors contributing to the lower severity observed in these regions. As the world gradually recovers from the COVID-19 pandemic, the varying impact it has had across different regions underscores our incomplete understanding of its trajectory and broader implications for global health. Despite its devastating global impact [1,2,3,4,5,6], the pandemic’s progression in Sub-Saharan Africa has been milder than expected, challenging initial forecasts and emphasizing the need for an in-depth investigation into the underlying factors [23]. Emerging hypotheses suggest several potential contributors to this trend, including the continent’s younger demographic profile, limited access to diagnostic tests, genetic factors, geographic settings, previous vaccinations, and lower rates of comorbidities [24]. Among these, the hypothesis that co-infections with parasitic diseases, including malaria and helminthiasis, stand out as particularly compelling [25,26,27]. Particularly, the interactions between helminths and the immune system present paradoxical scenarios during the COVID-19 pandemic. On the one hand, the immunomodulatory effects of these parasites could potentially reduce excessive inflammation, a characteristic of severe COVID-19 cases, thereby lessening symptom severity and lowering mortality rates [26]. This anti-inflammatory impact may have contributed to the observed milder course of COVID-19 in areas with high rates of helminth infections, offering natural resistance against the severe inflammatory reactions leading to severe COVID-19 [28]. On the other hand, the immune modulation induced by helminths might negatively affect vaccine efficacy [29]. This multifaceted scenario highlights the urgency for comprehensive studies that not only unravel the intricate interplay between infectious and parasitic diseases but also inform targeted health interventions and policy-making, crucial for improving resilience against future pandemics. Lymphatic Filariasis (LF), also known as elephantiasis and caused by parasitic worms such as *Wuchereria bancrofti*, *Brugia malayi*, and *Brugia timori*, is a prominent helminth infection affecting humans [30]. It is estimated that at least 36 million people suffer from chronic manifestations of LF, with over 15 million experiencing lymphoedema [31]. LF presents a significant public health challenge in tropical and subtropical areas, including Sub-Saharan Africa, where it causes chronic suffering and disability and imposes a heavy socio-economic burden on communities [31]. Like many helminths, LF parasites exert sophisticated immunoregulatory mechanisms to ensure their survival within the human host [32]. These mechanisms involve a complex interaction between the human immune system and LF parasites, with a notable strategy being the induction of a modified Th2 environment characterized by anti-inflammatory cytokines, such as IL-10 and TGF-β, which help downregulate pro-inflammatory responses and promote parasite tolerance. In addition, filarial nematodes, another important group to point out, are soil-transmitted helminths (STH), which are highly prevalent in (sub-) tropical areas, such as sub-Saharan Africa, mainly affecting regions with poor hygiene and sanitation standards [33]. STH infections, such as *Ascaris lumbricoides*, belong to the most common infections worldwide [33]. According to estimations is, almost one quarter of the world’s population (1.5 billion people) infected with soil-transmitted helminths [33]. STHs remain in the gastrointestinal tract, often leading to asymptomatic infections; nevertheless, a high worm burden, especially in children, can result in diarrhea, abdominal pain, and malnutrition [33]. The relationship between viral infections such as COVID-19 and chronic parasitic diseases remains poorly understood, making this research critical for illuminating the complex dynamics involved. In helminth endemic regions, including Ghana, where LF and COVID-19 coexist, there are unresolved questions about their combined effects on the body’s response to the coronavirus, especially in terms of vaccine effectiveness and disease progression. The present study aims to investigate the interaction between COVID-19 and LF within a unique cohort from Ghana, focusing on the humoral response following receival of the COVID-19 vaccine ChAdOx1-S (AstraZeneca). Moreover, various aspects of the complex interactions between COVID-19 and chronic filarial infections, including pathology and additional soil-transmitted seropositivity, were investigated in COVID-19-vaccinated individuals but also following a natural SARS-CoV-2 infection. This research is particularly significant given the coexistence of these infections in regions where both are endemic. By analyzing how LF may influence the immune response to COVID-19 and the efficacy of the vaccines, the study aims to uncover potential synergies or conflicts that could inform more tailored public health strategies.

## 2. Materials and Methods

### 2.1. Study Design and Clinical Characteristics of Participants

During the COVID-19 pandemic, blood samples of Lymphatic filariasis (LF) participants with Lymphoedema (LE) pathology (*n* = 222) were collected in the upper east region of Ghana (Kassena-Nankana District) in December 2021. Participants were enrolled from 12 sub-districts in the study, which was part of a larger and ongoing clinical trial called “Tackling the Obstacle to fight Filariasis and Podoconiosis” (TAKeOFF). Ethical approval was obtained by the local ethics committee (approval code: CHRPE/AP/525/17) and the ethical board of the University Hospital Bonn (approval code: 325/21 and 439/20). The controlled trial was registered under the following number: ISRCTN14042737 and the clinical outcome has been previously published by Batsa-Debrah et al. [34]. Sample collection was performed in Ghana, and samples were shipped to the Institute of Medical Microbiology, Immunology, and Parasitology (IMMIP) of the University Hospital of Bonn, where the experimental part was performed (Figure 1). A total of 222 participants (34 males and 188 females) were included in the study (Table 1). Treated lymphatic filariasis participants with lymphoedema pathology in different stages were involved in the study (Table 1). Participants were divided into two groups according to their lymphoedema (LE) stage: LE Stage 1–3 (Group A) and LE Stage 4–6 (Group B). Group A can be subdivided into two treatment arms: 100 mg/day of doxycycline, 200 mg/day of doxycycline, and a placebo group. Group B contained participants who received 200 mg/day of doxycycline and a placebo group. All participants received treatment for 6 weeks. Samples were taken 36 months post-treatment. Each participant gave informed consent for participation. The lymphoedema staging of the legs was performed at baseline and 36 months follow-up. The overall staging gives the highest achieved stage of a participant. An additional pregnancy test was performed for all female participants; however, all females were negative. The Filariasis Test Strip (FTS) (Alere Scarborough, Scarborough, ME, USA) for detection of circulating filarial antigen (CFA) was carried out 36 months after treatment for all participants. Due to limited COVID-19 test availabilities and capacities, none of the participants (*n* = 222) had ever tested positive for SARS-CoV-2 by PCR or rapid antigen test (Table 1). However, COVID-19-related symptoms between December 2020 and December 2021 were well documented using a questionnaire. The following symptoms were included: loss of appetite, loss of odor, fever, headache, cough, common cold, sore throat, shortness of breath, respiratory problems, fatigue, sweating or chills, musculoskeletal pain, abdominal pain, nausea/vomiting, and tightness in the chest. Furthermore, the presence of experienced symptoms, onset, and duration were also recorded. The majority (84%) of participants did not exhibit any COVID-19-related symptoms (Table 1), and none of the participants was hospitalized between December 2020 and December 2021 due to severe COVID-19 or any other clinical condition. All participants received the vector-based COVID-19 vaccine ChAdOx1-S (Vaxzevria) developed by the University of Oxford and AstraZeneca. Participants were either grouped into incompletely (1. dose) or completely (2. doses) vaccinated. Information on vaccine administration was obtained by providing participants with vaccine cards. All vaccine doses of the LF participants were administered between October 2021 and November 2021. In addition, blood samples of 12 healthy endemic controls (HEC) were collected in December 2021. The majority (*n* = 10) of HECs received a complete COVID-19 vaccination (between March 2021 and August 2021). All HECs were tested CFA negative.

### 2.2. Quantification of SARS-CoV-2-Specific Antibodies

SARS-CoV-2-Spike-specific IgA (Catalog number: EI2606-9601A) and IgG (Catalog number: EI2606-9601G) and Nucleocapsid-specific IgG (Catalog number: EI2606-9601-2G) were quantified in the plasma of Ghanaian lymphatic filariasis infected individuals using enzyme-linked immunosorbent assay (ELISA) kits (Euroimmun, Lübeck, Germany). The assays were performed automatically using the predesigned program of the analyzer I automate (Euroimmun, Lübeck, Germany). 100 µL/well of prediluted samples (1:101), calibrator, and controls were added to the pre-coated 96-well plate and incubated for 60 min at 37 °C. After washing (3 × 300 µL/well), 100 µL/well conjugate (peroxidase-labeled anti-human IgG) was added and incubated for 30 min at RT. After another washing step, 100 µL/well substrate solution was added and incubated for 15 min at RT. Finally, 100 µL/well stop solution was added, and optical densities (OD) were measured at 450 nm. The extinction of the calibrator corresponds to the cut-off value, and the ratio corresponds to the sample’s extinction in relation to the calibrator, allowing a semiquantitative measurement of participant samples. The ratio was calculated according to the manufacturer’s protocol using the following formula: (extinction of sample/extinction of the calibrator) = ratio. Ratios <0.8 were considered negative, 0.8–1.1 as equivocal, and >1.1 as positive.

### 2.3. Quantification of Ascaris Lumbricoides Specific IgG

*Ascaris lumbricoides*-specific IgG (Catalog number: EIA-3817) were quantified in the plasma of Ghanaian lymphatic filariasis infected individuals using enzyme-linked immunosorbent assay (ELISA) kits (DRG Instruments GmbH, Marburg, Germany). The pre-diluted (1:101) samples, controls, and standards were added (100 µL/well) to the pre-coated 96-well plate and incubated for 1 h at 37 °C. After washing (3 × 300 µL/well), the conjugate was added (100 µL/well) and incubated for 30 min at RT. After another washing step, tetramethylbenzidine (TMB) substrate was added (100 µL/well) and incubated for 15 min at RT. Finally, a stop solution was added (100 µL/well), and the extinction was measured at wavelengths of 450/620 nm. The validation criteria of the assay were thoroughly verified according to the manufacturer’s instructions. The absorbance was converted into units according to the manufacturer’s recommendation using the following formula (sample absorbance × 10/cut-off) = DRG units = DU. Results were interpreted as follows: >11 DU as positive, 9–11 DU as equivocal, and <9 DU as negative. The diagnostic specificity, defined as the probability of scoring negative in the absence of the specific analyte, is given at 95%. The sensitivity, defined as the probability of scoring positive in the presence of the specific analyte, is given at 100%. A cross-reaction with antibodies against *Toxocara canis*, *Trichinella*, *Fasciola*, *Filaria*, and *Strongyloides* cannot be excluded.

### 2.4. Quantification of Human Filarial-Specific Antibodies

*Acanthocheilonema viteae* ELISA-Kit (Bordier Affinity Products, Crissier, Switzerland, Catalog number: 9400) was used for the quantitative detection of IgG immunoglobulins specific for human filarial infections, including lymphatic filariasis (caused by *Wuchereria bancrofti* and *Brugia malayi*), loaosis, onchocerciasis, and mansonelliasis in plasma of Ghanaian LE individuals, according to the manufacturer’s protocol. Briefly, the pre-coated 96-well plate was blocked with 300 µL buffer for 15 min. at RT. The pre-diluted (1:201) samples and controls were added (100 µL/well) and incubated for 30 min at 37 °C. After washing (4 × 300 µL/well), the conjugate was added (100 µL/well) and incubated for 30 min at 37 °C. After another washing step, the substrate solution was added (100 µL/well) and incubated for 30 min at 37 °C, followed by a stop solution (100 µL/well) to terminate the reaction. The absorbance was measured at a wavelength of 450 nm. The validation criteria were thoroughly verified according to the manufacturer’s instructions. The sample absorbance was converted into index values with the following formula: (sample absorbance value/absorbance cut-off control) = Index. Results were interpreted as follows: >1.0 Index as positive and <1.0 as negative. The diagnostic specificity and sensitivity are given at 98% and 95%, respectively. Cross-reactivity occurred with ascariasis, trichinellosis, ancylostomiasis, fasciolosis, and cystic echinococcosis.

### 2.5. Quantification of SARS-CoV-2 Neutralizing Antibody Levels

SARS-CoV-2 neutralizing antibodies were quantified using a competitive multiplex immunoassay (SARS-CoV-2 Variants Neutralizing Antibody 6-plex ProcartaPlex Panel) manufactured by Thermo Fisher (Waltham, MA, USA, Catalog number: EPX060-16018-901). The assay allows the comparison of variant-specific neutralizing antibody capacity towards six SARS-CoV-2 variants, including Wildtype (WT), Alpha (B.1.1.7), Beta (B.1.351), Gamma (P.1), Delta (B.1.617.2) and Omicron (B.1.1.529). 50 µL of magnetic capture beads were added to each well of the provided 96-well plate and washed with 150 µL 1X washing solution. 25 µL of prediluted samples (1:100) were added, followed by 25 µL assay diluent. Positive and negative control was prepared according to the manufacturer’s recommendation and added to the dedicated wells. Samples were incubated for 2 h at RT while shaking (500 rpm/min). After two washing steps, 25 µL of prediluted 1X detection antibody was added to each well and incubated for 30 min at RT while shaking. After another two washing steps, 50 µL of Streptavidin-PE solution was added to each well and incubated for another 30 min at RT on a plate shaker. After two more washing steps, 120 µL of reading buffer was added to each well, and the plate was incubated for 5 min while shaking before the plate was measured using the MagPix Luminex instrument. The results were analyzed using the following neutralization equation: (1—(MFI of samples/MFI of negative control)) × 100 = Neutralization (%).

### 2.6. Quantification of Systemic Cytokine and Chemokine Levels

Cytokine storm 21-plex human ProcartaPlex Panel (Thermo Fisher, Waltham, MA, USA, Catalog number: EPX210-15850-901) was used to quantify systemic cytokines and chemokines in the plasma of Ghanaian Lymphoedema participants. The assay allowed measurement of the following analytes: G-CSF (CSF-3), GM-CSF, IFN alpha, IFNγ, IL-1 beta, IL-2, IL-4, IL-5, IL-6, IL-8 (CXCL8), IL-10, IL-12p70, IL-13, IL-17A (CTLA-8), IL-18, IP-10 (CXCL10), MCP-1 (CCL2), MIP-1 alpha (CCL3), MIP-1 beta (CCL4), TNF alpha, and TNF beta. All previously mentioned analytes are related to the cytokine release syndrome (CRS). The assay was carried out manually according to the manufacturer’s instructions by adding magnetic capture beads (50 µL/well) to the designated wells on the provided 96-well plate. After washing (1 × 150 µL/well), undiluted samples were added (25 µL/well), followed by assay diluent (25 µL/well). Standard and controls were prepared according to the manufacturer’s recommendation, and samples were incubated for 2 h at RT while shaking (500 rpm/min). After two washing steps, 25 µL of 1× Detection antibody was added to each well and incubated 30 min. at RT while shaking. After another two washing steps, 50 µL of Streptavidin-PE solution was added to each well and incubated for another 30 min. at RT on a plate shaker. After the final two washing steps, 120 µL of reading buffer was added to each well, and the plate was incubated for 5 min. The plate was then measured using the MagPix Luminex instrument, and the results were analyzed using the ProcartaPlex analyst application (Thermo Fisher).

### 2.7. Statistics

Data were analyzed using GraphPad Prism (version 10.4.1, La Jolla, CA, USA). We described the characteristics of the study participants using mean values ± (SD), Minimum, and Maximum for continuous variables. The values of categorical variables were presented as frequency distribution with percentages. The nonparametric Kruskal-Wallis test, followed by Dunn’s posthoc test, was used for group comparison along with the Mann-Whitney U-test. A *p*-value < 0.05 was considered statistically significant.

## 3. Results

### 3.1. High SARS-CoV-2 Seroprevalence Among COVID-19 Unvaccinated and Vaccinated Individuals in Ghana

Since the circulation of SARS-CoV-2 in African countries is not well-documented due to the lack of testing capacity and the high prevalence of asymptomatic infections [24], we first evaluated Nucleocapsid (NCP) and Spike (S) protein seroprevalences to discriminate between vaccine-induced and infection-induced SARS-CoV-2 antibody expressions. The analysis of the unvaccinated cohort (*n* = 141) revealed that a remarkable portion had serological evidence of exposure to SARS-CoV-2: 56% had detectable levels of Spike-specific IgA, and 39% were positive for Spike-specific IgG, as illustrated in Figure 2A,B, respectively. These results suggest a considerable degree of previous exposure to the virus among the unvaccinated individuals. Further investigation into the immune responses of vaccinated participants involved a SARS-CoV-2 Nucleocapsid protein-specific (NCP) IgG ELISA. This assay was instrumental in distinguishing between vaccine-induced immunity and immunity developed from natural infection. Our findings showed that among the incompletely vaccinated individuals (*n* = 63), 31.7% tested positive for NCP-specific IgG, indicating a likelihood of previous viral infection (Figure 2C). In contrast, the fully vaccinated group (*n* = 18) demonstrated a higher NCP-seropositivity rate of 44.4% (Figure 2D). These observed rates of seropositivity to the NCP antigen in vaccinated individuals reflect additional viral encounters following vaccination. The disparity in NCP-seropositivity between those with incomplete and complete vaccination courses suggests a complex interplay between vaccination-induced immunity and natural infection.

### 3.2. Robust SARS-CoV-2-Specific Antibody Response in Lymphatic Filariasis Patients After Incomplete and Complete COVID-19 Vaccination

After confirming a high SARS-CoV-2 seroprevalence in the LF cohort, both among vaccinated and unvaccinated individuals, we first investigated how LF pathology stages influence antibody response to COVID-19 vaccination. The results indicated that the Lymphoedema stages of the participants did not influence the SARS-CoV-2 antibody response after vaccination (Appendix A). In the next step, we compared the SARS-CoV-2-specific antibody levels among lymphatic filariasis (LF) infected individuals and completely COVID-19-vaccinated healthy endemic controls (HEC). The LF participants were further grouped according to their COVID-19 vaccination and SARS-CoV-2 serostatus into four groups: (1) COVID-19 unvaccinated and SARS-CoV-2 seronegative, (2) COVID-19 unvaccinated but SARS-CoV-2 seropositive, (3) incompletely (1. dose) COVID-19 vaccinated and (4) completely (2. doses) COVID-19 vaccinated. The earlier vaccination date (between March 2021 and August 2021) of the HECs complicates the comparison to LF participants who received their vaccination shortly before sampling (between October and November 2021). Despite these discrepancies within the vaccination date, the HECs elevated antibody levels compared to the COVID-19 unvaccinated and SARS-CoV-2 seronegative LF group. No statistical differences were observed among COVID-19-vaccinated LF groups and HECs. By further comparing the four LF groups, it was shown that the median levels of SARS-CoV-2-specific IgA (Figure 3A) and IgG (Figure 3B) were highest in the completely vaccinated group. The incompletely vaccinated group also showed increased IgA levels, although there was considerable variability within this group. Both vaccinated groups demonstrated notably higher IgA and IgG levels compared to the unvaccinated seronegative and seropositive groups. These findings highlight that COVID-19 vaccination induces a robust SARS-CoV-2-specific antibody response in LF patients. To further characterize the effect of helminth infections on the COVID-19 vaccination response, we analyzed SARS-CoV-2 IgA and IgG expressions related to their parasitic status. The parasitic status was grouped according to Negative, Single Positive, or Double Positive for the Circulating filarial antigen (CFA) test, *Ascaris lumbricoides* IgG, and *Acanthocheilonema viteae* IgG. The data indicated that none of the helminths remarkably affected COVID-19-mediated antibody induction. Among single and double-positive participants, enhanced antibody responses were observed in the incompletely and completely vaccinated groups (Appendix A). Furthermore, the potential effect of the parasitic status on the SARS-CoV-2 antibody response and the possible impact on the systemic cytokine and chemokine profile was further analyzed. However, no effect has been observed, and comparable levels of Th1, Th2, Th17, and Treg-related cytokines were detected within the plasma of lymphatic filariasis individuals, irrespective of their filarial status (Appendix A).

### 3.3. Increased Neutralizing Antibody Levels Towards Variants of Concern in COVID-19 Vaccinated Filaria Seropositive Individuals

Since antibody production may not necessarily translate into neutralizing potential, we compared the levels of SARS-CoV-2 neutralizing antibodies in lymphatic filariasis (LF) infected individuals and completely COVID-19 vaccinated healthy endemic controls (HEC). The LF participants were grouped as previously described into: (1) COVID-19 unvaccinated and SARS-CoV-2 seronegative, (2) COVID-19 unvaccinated but SARS-CoV-2 seropositive, (3) incompletely (1. dose) COVID-19 vaccinated and (4) completely (2. doses) COVID-19 vaccinated. Despite deviations within the vaccination date of LF participants and HECs (March-August 2021. October-November 2021, respectively), the HECs elevated neutralizing antibody levels compared to the COVID-19 unvaccinated and SARS-CoV-2 seronegative LF group. No statistical differences were observed among HECs and completely COVID-19-vaccinated LF participants. By further comparing the four subdivided LF groups, it was observed that the neutralizing capacity of SARS-CoV-2 specific antibodies within the incomplete (*n* = 63) and complete (*n* = 18) COVID-19 vaccinated lymphatic filariasis participants was notably higher compared to the unvaccinated and SARS-CoV-2 seronegative control group (*n* = 37) (Figure 4). By comparing the neutralizing antibody capacity towards six SARS-CoV-2 variants including Wildtype (WT), Alpha (B.1.1.7), Beta (B.1.351), Gamma (P.1), Delta (B.1.617.2) and Omicron (B.1.1.529), higher levels towards the different variants of concern have been observed in the COVID-19 vaccinated groups, except for the Omicron variant where just complete COVID-19 vaccinated individuals higher levels achieved (Figure 4F). However, the COVID-19 unvaccinated but SARS-CoV-2 seropositive group (*n* = 48) showed reduced levels of neutralizing antibodies compared to the vaccinated groups. Interestingly, the SARS-CoV-2 seropositive group had increased neutralizing potential compared to the unvaccinated control group, but only for the Wild type, Alpha, and Gamma variants (Figure 4). In the next step, the influence of an additional COVID-19 infection within the COVID-19 vaccinated groups on the neutralizing capacity of SARS-CoV-2 specific antibodies was analyzed, nevertheless, no differences have been observed (Appendix A).

### 3.4. Diminished SARS-CoV-2-Specific IgA Response in Ascaris Seropositive Individuals

After observation of a robust antibody response and high neutralizing potential following COVID-19 vaccination, a potential nematode effect on COVID-19 infection-induced responses was evaluated additionally. Therefore, the effect of *Ascaris lumbricoides* seropositivity on the SARS-CoV-2-Spike-specific IgA/IgG response within COVID-19-infected individuals was analyzed. In regard to IgA expression, our findings indicated a decrease (*p* = 0.0264) in SARS-CoV-2 specific IgA levels among individuals seropositive for *Ascaris lumbricoides* (*Ascaris*+) (Figure 5A). The median IgA levels were notably lower (39%) in the *Ascaris*+ group (*n* = 28) compared to the seronegative (*n* = 35) (*Ascaris*-) cohort, underscoring the impact of nematode infection on the humoral immune response to SARS-CoV-2. Concerning the SARS-CoV-2 IgG response, no difference was observed in IgG antibody levels between the *Ascaris* seronegative and positive groups (*p* = 0.1463) (Figure 5B). This suggests that nematode infection status does not markedly influence the IgG-mediated immune response against SARS-CoV-2. Overall, the data suggests that *Ascaris lumbricoides* infection is associated with a reduced IgA antibody response to SARS-CoV-2, yet it appears to have no crucial effect on the IgG antibody response. Due to the negative effect of filarial seropositivity on the COVID-19 infection-induced SARS-CoV-2 specific IgA antibody response, the possible negative effect on the neutralizing capacity of the SARS-CoV-2 specific antibodies towards six different variants of concern was analyzed. However, comparable neutralization potentials of SARS-CoV-2 antibodies were observed in the *Ascaris lumbricoides* seropositive group towards all six different VoCs (Figure 5C–H).

### 3.5. No Influence of Doxycycline Treatment 36-Months Before Blood Sampling of Lymphatic Filariasis Individuals with Lymphoedema Pathology Compared to the Placebo Control Group

Since all participants were part of a clinical trial and received a six-week doxycycline treatment (100 mg/day or 200 mg/day) 36 months before blood sampling, a potential effect of treatment on the SARS-CoV-2 specific antibody response post-COVID-19 infection and/or vaccination was analyzed. However, no statistical differences were observed for the SARS-CoV-2 specific IgA and IgG response among the two doxycycline treatment arms (100 mg/day and 200 mg/day) compared to the placebo control group within unvaccinated, incompletely and completely COVID-19 vaccinated individuals (Figure 6).

## 4. Discussion

Vaccines are one of the most successful medical inventions, crucial for eradicating or controlling common and fatal diseases [35]. For COVID-19, the early rollout of various vaccines significantly contributed to halting the pandemic [36,37,38]. Several studies have shown that helminths can modulate immune responses to vaccines [39,40,41]. Given the prevalence of helminth infections globally, especially in sub-Saharan Africa [33], evaluating COVID-19 vaccine efficacy in helminth-infected populations is essential for optimizing vaccination programs. In this context, the present study explored the humoral immune response following COVID-19 vaccination with ChAdOx1-S (AstraZeneca) in the lymphatic filariasis (LF) cohort in Ghana. Our initial seroprevalence analyses revealed a remarkable portion of the unvaccinated cohort with serological evidence of exposure to SARS-CoV-2: 56% had detectable levels of Spike-specific IgA, and 39% were positive for spike-specific IgG. These high seroprevalence rates could be due to asymptomatic infections and limited testing capacity in the region. Our data are comparable with the results of previous prevalence analyses in Africa [42,43]. Here, we uniquely employed Nucleocapsid (NCP) and Spike (S) protein seroprevalences to differentiate between vaccine-induced and infection-induced antibodies. Our findings imply a considerable degree of previous exposure to the virus among both vaccinated and unvaccinated individuals, which has important implications for public health strategies and our understanding of SARS-CoV-2 spread in similar contexts. Our study found that COVID-19 vaccination induces a robust SARS-CoV-2-specific antibody response in LF patients. The median levels of SARS-CoV-2-specific IgA and IgG were highest in the completely vaccinated group, highlighting the robustness of the antibody response in these individuals. The incompletely vaccinated group also showed increased IgA levels, although with notable variability. Moreover, our results suggest that LF pathology stages do not influence the SARS-CoV-2 antibody response after vaccination. This finding is important for vaccination strategies in populations with similar health profiles, indicating that COVID-19 vaccines remain effective regardless of LF pathology. We analyzed the impact of circulating filarial antigen (CFA) and filarial serological positivity as well as seropositivity towards other helminth infections (*Ascaris lumbricoides*) on antibody responses to COVID-19 vaccines and found that these did not considerably affect antibody induction post-COVID-19 vaccination. This suggests that co-infections with these helminths do not pose a barrier to achieving effective immunization against COVID-19. These findings are relevant for understanding the broader implications for co-infected populations and support the continued rollout of COVID-19 vaccines in helminth-endemic regions. A plausible explanation for the lack of crucial impact of helminth infections on the immune response to the ChAdOx1-S COVID-19 vaccine could be the nature of the vaccine itself. AstraZeneca’s vaccine, also known as ChAdOx1-S or Vaxzevria, uses a viral vector based on a weakened version of an adenovirus that causes infections in chimpanzees [44]. This viral vector approach may induce a strong and broad immune response that is less susceptible to modulation by helminth-induced immune changes. Helminths are known to induce a modified type 2 immune response (Th2) characterized by the production of cytokines such as IL-4, IL-5, IL-13, and IL-10, which can potentially modulate vaccine responses [39]. However, the ChAdOx1-S vaccine is designed to stimulate a robust type 1 immune response (Th1) with high levels of IFN-γ and TNF-α, which are crucial for antiviral immunity [45]. The strong Th1 response induced by the vaccine might override the modulatory effects of helminths, resulting in effective immunization. Several potential mechanisms might explain why AstraZeneca’s vaccine remains effective in the presence of helminth infections: The use of a viral vector can generate strong immune responses, possibly overcoming any immunomodulatory effects induced by helminths [39,44,45]. The ability of the viral vector to present multiple epitopes could stimulate a broad and diverse immune response, making it more resilient to immune modulation by helminths [46]. Our findings are particularly encouraging for communities in helminth-endemic regions. They suggest that the COVID-19 vaccination programs using AstraZeneca’s vaccine can be rolled out without the need for extensive deworming campaigns beforehand. This is crucial for maintaining high vaccination coverage and protecting vulnerable populations from COVID-19, especially in areas where healthcare resources are limited and helminth infections are prevalent. Another interesting aspect of our data is the robust antibody expression and neutralizing potential against different SARS-CoV-2 variants in completely vaccinated individuals. Among the other groups, only completely vaccinated individuals showed increased levels of neutralizing antibodies towards the Omicron variant, indicating no circulation or exposure to this variant of concern at the sampling time point. While relatively high neutralizing antibody levels are seen in incompletely vaccinated individuals, a huge variability in antibody production and neutralizing potential is seen, suggesting that some individuals in this group may not be sufficiently protected against severe forms of infection. This finding aligns with previous literature on the topic [47,48] and confirms the necessity to adopt a full vaccination regime. Our findings also indicated no differences in SARS-CoV-2 antibody responses among participants who received doxycycline treatment 36 months before blood sampling compared to the placebo control group. Doxycycline, in addition to its indirect macrofilaricidal capacity, leading to sterility and death of adult worms [49,50], was also shown to have various immunomodulatory effects through inhibition of metalloproteinases and was suggested as a potential treatment in hospitalized COVID-19 patients [51]. These immunomodulatory effects can potentially affect immune responses to COVID-19 vaccines. To exclude any direct effect of doxycycline on the immune responses to COVID-19 vaccines, we compared SARS-CoV-2 antibody levels in doxycycline-treated and placebo patients. Our data indicates that prior doxycycline treatment has no impact on vaccine-induced immunity. This result is relevant for treatment protocols and their potential impact on vaccine-induced immunity, supporting the idea that prior doxycycline treatment can be safely administered without compromising vaccine efficacy. The main limitation of this work is its regional scope in Ghana, which may limit the generalizability of the findings to other regions with different epidemiological profiles. Further studies are needed to confirm these findings in other geographic and demographic contexts. Additionally, despite controlling for many variables, there might still be unaccounted confounders that could have influenced the results. Factors such as variations in individual immune responses, environmental influences, and genetic diversity could potentially impact the observed outcomes. Moreover, the lack of PCR testing for COVID-19 in our cohort could mean that some SARS-CoV-2 exposures were potentially underestimated, as some infected individuals may not produce antibodies [52,53]. Moreover, the relative rapid waning of antibodies after a virus encounter may have led to the underestimation of asymptomatic individuals [54]. PCR testing was crucial for identifying active infections and understanding the true spread of the virus. In Ghana and other parts of West Africa, limited access to PCR testing has been a significant challenge in accurately tracking COVID-19 cases [55]. Previously mentioned factors complicate the evaluation of vaccine efficacy in terms of incidence or clinical presentation. However, the documentation of COVID-19-related symptoms revealed that only a minority of study participants (16%) exhibited any flu-like symptoms, and no participant in our study cohort suffered from severe COVID-19 related to hospital admission. Potential beneficial or protective measures against severe COVID-19, such as additional vaccines (e.g., BCG or influenza), were also considered but did not show any impact on the obtained data and were not evaluable regarding disease severity due to the absence of severe or hospitalized COVID-19 cases within the present work. The neutralization assay, which was used for the detection of variant-specific neutralizing antibodies, is a cell- and virus-free ACE2 competition assay, allowing a multiplex readout that can be performed under biosafety level 2 conditions. In regard to the use of antibodies to detect helminth infections, while serological analyses have the potential to detect current and previous helminth infections, which extends the validity and robustness of our conclusions, they may also introduce variability in the detection accuracy and sensitivity. While our study cohort includes only ChAdOx1-S vaccinated individuals, it is important to verify if our findings are confirmed with other vaccines. This was not possible in the present study due to the low COVID-19 vaccine coverage in the region [56], the logistic challenges posed by mRNA-based vaccines in sub-Saharan Africa, but also the high vaccine hesitancy, particularly towards mRNA vaccines in the region [56]. Another limitation was the involvement of healthy endemic controls and their discrepancies within the vaccination time frame, which complicated the comparison between LF participants and controls. Despite the fact that all healthy controls were tested negative for lymphatic filariasis, the presence of undetected co-infections and the undeterminable infection history further complicated the analysis, especially by using serological tests for the detection of helminth infections. This further highlights the complications within the evaluation of vaccine efficacy and the complex interplay of contributing factors such as pre-existing infections and co-infections. Furthermore, the present study contained a gender bias, which is highly skewing towards women. There was no gender effect observed in the present data, which is consistent with other studies where age was determined as a dominating factor in vaccine-induced responses irrespective of gender [21]. In addition to this, the present study also contained a pathology bias, which is skewing towards those with a lower pathology degree (lymphoedema stage 1–3), and despite the fact that the stage did not show any influence in the present work, a higher sample size of participants with increased pathology might lead to deviations from our obtained data. To our knowledge, there is currently no literature available on whether the extent of LF pathology has an influence on humoral vaccine-induced responses, neither for the COVID-19 vaccines nor for other vaccines. Further investigations are required to fill this knowledge gap. In addition, while our study focused on humoral immune responses to COVID-19 vaccines, due to logistical challenges, T cell responses might also play a key role [57].

## 5. Conclusions

Altogether, the present retrospective study provides insight into the ability of filarial seropositivity and/or co-infection to modify COVID-19 vaccine efficacy in humans. It can be stated that an adenovirus-vectored COVID-19 vaccine in lymphatic filariasis individuals induced a strong humoral immune response. This vaccine-induced response was independent of the lymphoedema stage and serological parasitic status. Moreover, notably elevated neutralizing antibody levels towards six different SARS-CoV-2 variants of concern were observed within the COVID-19 vaccinated groups independent of filarial status. These findings support the continued rollout of COVID-19 vaccines in helminth-endemic regions, emphasizing the need for comprehensive public health strategies to manage co-infections and improve vaccine coverage. The manuscript provides valuable data and insights into the complex interplay between helminth infections and COVID-19, with important implications for public health policies in endemic regions. However, the identified limitations highlight areas for further research and validation to enhance the applicability of these findings across diverse populations.

## Figures and Tables

**Figure 1 vaccines-13-00312-f001:**
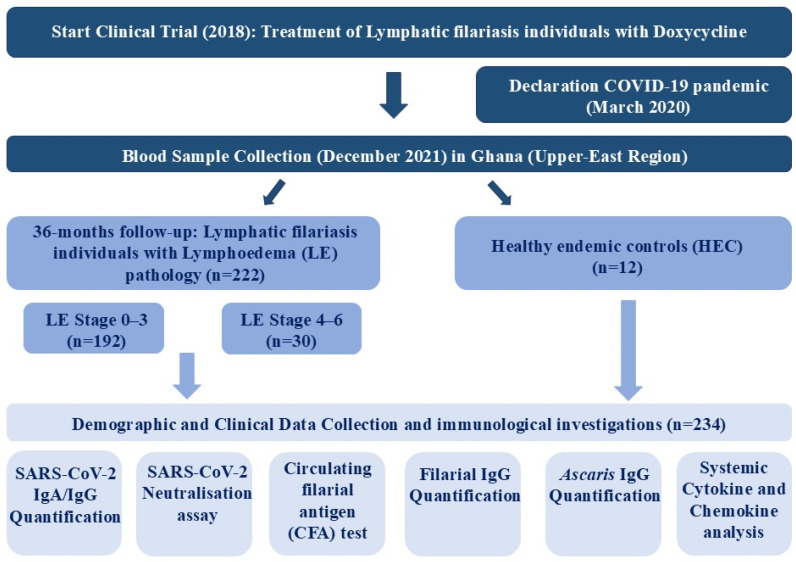
Flowchart of the study design and timeline for assessing the immunological responses in lymphatic filariasis patients treated with doxycycline. Blood samples were collected in December 2021 during the COVID-19 pandemic from individuals with lymphatic filariasis who were part of a treatment trial (started in 2018) in the Upper-East Region of Ghana. The study included 222 individuals with lymphoedema (LE) pathology, divided into two groups based on LE stage (0–3 and 4–6), along with 12 healthy endemic controls (HEC). Demographic and clinical information was collected, as well as immunological investigations, including SARS-CoV-2 IgA/IgG quantification, SARS-CoV-2 neutralization assay, circulating filarial antigen (CFA) test, filarial IgG quantification, Ascaris IgG quantification, and systemic cytokine and chemokine analysis. Additionally, blood samples from 12 healthy endemic controls were collected. Sample and data collection occurred in Ghana, while immunological investigations (except for the CFA test) were conducted at the Institute for Medical Microbiology, Immunology, and Parasitology (IMMIP) in Bonn, Germany.

**Figure 2 vaccines-13-00312-f002:**
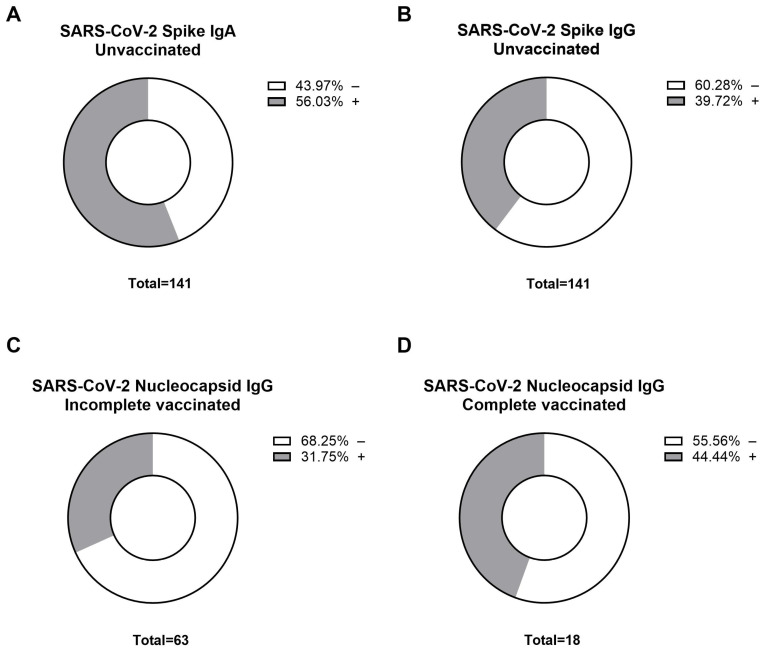
Seropositivity rates for SARS-CoV-2-specific antibodies between unvaccinated and vaccinated cohorts in Ghana. Donut charts represent the seroprevalence of (**A**) Spike-specific IgA antibodies in the unvaccinated cohort (*n* = 141), with 56.03% seropositive (grey) and 43.97% seronegative (white); (**B**) Spike-specific IgG antibodies in the unvaccinated cohort (*n* = 141), with 39.72% seropositive (grey) and 60.28% seronegative (white); (**C**) Nucleocapsid protein-specific (NCP) IgG antibodies in the partially vaccinated group (*n* = 63), with 31.75% seropositive (grey) and 68.25% seronegative (white); (**D**) Nucleocapsid protein-specific (NCP) IgG antibodies in the fully vaccinated group (*n* = 18), with 44.44% seropositive (grey) and 55.56% seronegative (white).

**Figure 3 vaccines-13-00312-f003:**
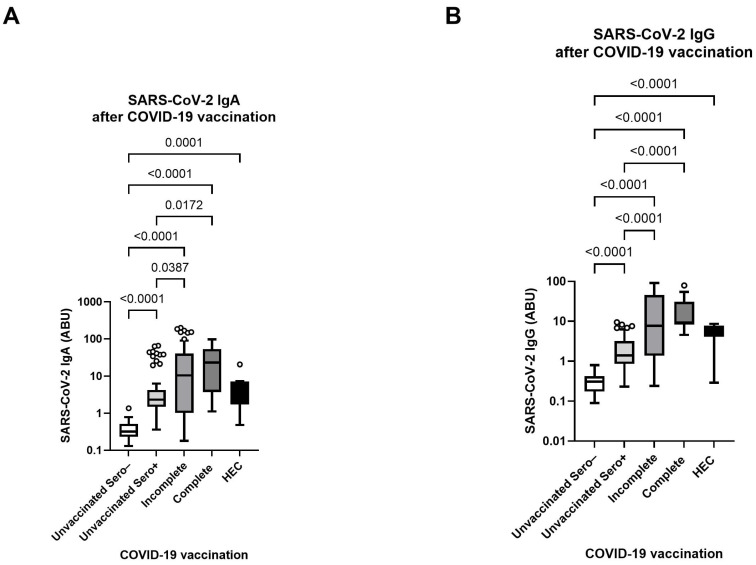
SARS-CoV-2-specific antibody levels in different vaccination and serostatus groups. Box plots represent SARS-CoV-2 spike-specific IgA (**A**) and IgG (**B**) antibody expressions in Ghanaian lymphatic filariasis (LF) infected individuals with varying COVID-19 vaccination status compared to completely vaccinated healthy endemic controls (HEC) (*n* = 10) (black). LF participants were grouped into COVID-19 unvaccinated and SARS-CoV-2 seronegative (*n* = 37) (white), COVID-19 unvaccinated and SARS-CoV-2 seropositive (*n* = 87) (light grey), incompletely (*n* = 63) (medium grey) and completely (*n* = 18) COVID-19 vaccinated (dark grey). Indicated *p* values were calculated using Kruskal-Wallis’ test followed by Dunn’s comparison post hoc to compare all groups. Bars represent the median ± IQR of antibody binding units (ABU). Significance is accepted if *p* < 0.05.

**Figure 4 vaccines-13-00312-f004:**
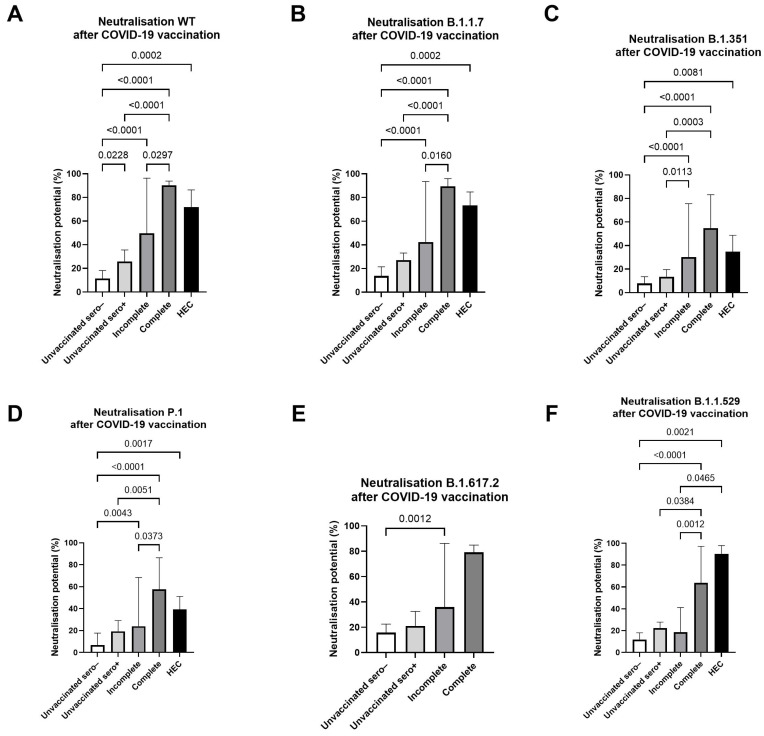
Neutralization potential against different SARS-CoV-2 variants in various vaccination and serostatus groups. Bars represent the expression of SARS-CoV-2 specific neutralizing antibodies in Ghanaian lymphatic filariasis (LF) infected individuals with varying COVID-19 vaccination status compared to completely vaccinated healthy endemic controls (HEC) (*n* = 10) (black). LF participants were grouped into COVID-19 unvaccinated and SARS-CoV-2 seronegative (white), COVID-19 unvaccinated and SARS-CoV-2 seropositive (light grey), incompletely (1.dose) COVID-19 vaccinated (medium grey) and completely (2.doses) COVID-19 vaccinated (dark grey). The neutralizing potential towards six major SARS-CoV-2 variants, including Wildtype (WT) (**A**), Alpha (B.1.1.7) (**B**), Beta (B.1.351) (**C**), Gamma (P.1) (**D**), Delta (B.1.617.2) (**E**) and Omicron (B.1.1.529) (**F**) was compared. Indicated *p* values were calculated using Kruskal-Wallis’ test followed by Dunn’s comparison post hoc to compare all groups. Bars represent the median ± IQR of antibody neutralizing potential (%). Significance is accepted if *p* < 0.05.

**Figure 5 vaccines-13-00312-f005:**
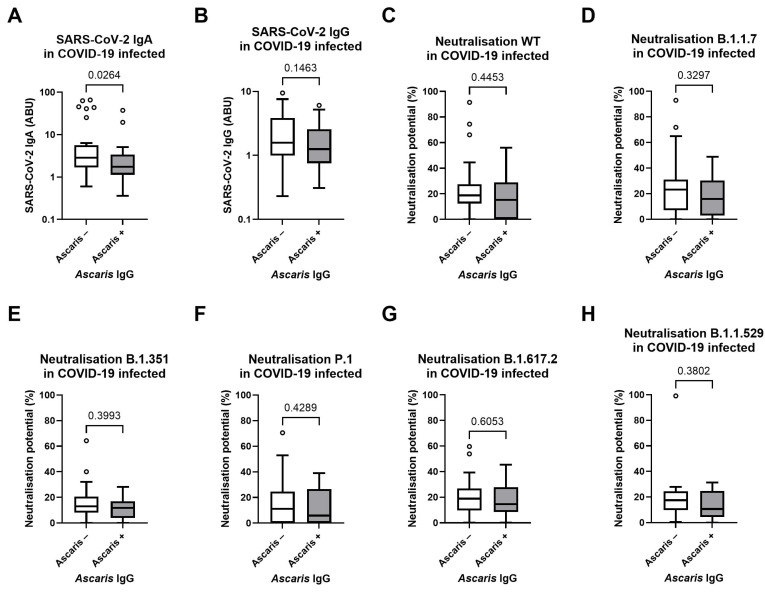
Influence of *Ascaris lumbricoides* IgG seropositivity on COVID-19 infection-induced SARS-CoV-2 Spike-specific antibody response and neutralizing potential. Reduced SARS-CoV-2 IgA levels were seen in the *Ascaris* positive (*n* = 28) group (grey) compared to the negative (*n* = 35) group (white) (**A**), as well as comparable SARS-CoV-2 IgG levels (**B**). Comparable neutralization capacity of SARS-CoV-2 specific antibodies towards six major SARS-CoV-2 variants within *Ascaris* seropositive COVID-19 infected participants compared to seronegative group (**C**–**H**). Indicated *p* values were calculated using the Mann-Whitney-U test. Bars represent the median ± IQR of antibody binding units (ABU) (**A** + **B**) and antibody neutralizing potential (%) (**C**–**H**). Significance is accepted if *p* < 0.05.

**Figure 6 vaccines-13-00312-f006:**
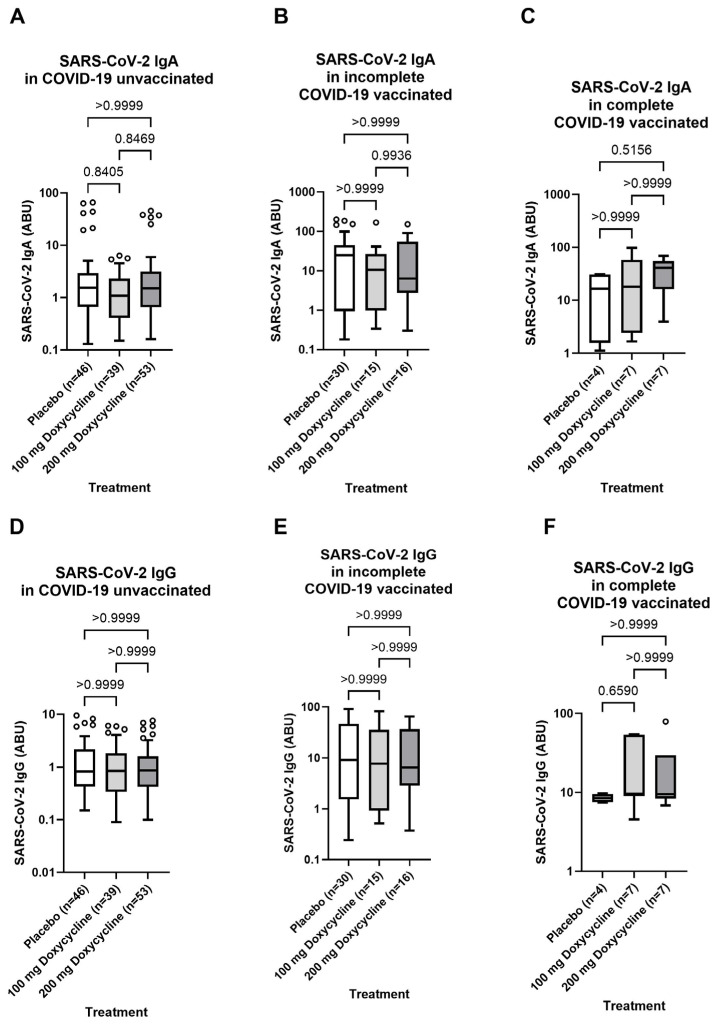
No effect of Doxycycline Treatment 36 months before blood sampling on infection- or vaccination-induced SARS-CoV-2 specific antibody response of Lymphatic filariasis participants. Comparable SARS-CoV-2 Spike-specific IgA (**A**–**C**) and IgG (**D**–**F**) antibody levels among unvaccinated (*n* = 141) (**A** + **D**), incomplete (*n* = 63) (**B** + **E**) and complete (*n* = 18) (**C** + **F**) COVID-19 vaccinated lymphatic filariasis individuals irrespective of a six-week treatment with 100 mg/day (light grey) and 200 mg/day (grey) Doxycycline 36-months before blood sampling compared to the Placebo Control group (white). Indicated *p* values were calculated using Kruskal-Wallis’ test followed by Dunn’s comparison post hoc to compare all groups. Bars represent the median ± IQR of antibody binding units (ABU). Significance is accepted if *p* < 0.05.

**Table 1 vaccines-13-00312-t001:** Demographic and clinical characteristics of study participants.

Sample Size (*n*=)	222
Age (Mean ± SD, Min–Max)	46.57 ± 9.21 (17–64)
Sex (M/F)	34 (15%)/188 (85%)
BMI (Min-Max, Mean ± SD)	11.49–42.44 (23.32 ± 4.05)
LE Staging (0/1/2/3/4/5/6)	2/5/130/55/0/1/29 0.9%/2.3%/58.6%/24.8%/0%/0.5%/13.1%
Treatment Group (A/B)	198 (89%)/24 (11%)
COVID-19 Positive test (No/Yes)	222 (100%)/0
COVID-19 Vaccination (No/1. Dose/2. Dose)	141 (63%)/63 (28%)/18 (8%)
BCG vaccination (No/Yes/Unknown)	65 (29%)/139 (63%)/18 (8%)
Influenza vaccination (No/Yes/Unknown)	65 (29%)/139 (63%)/18 (8%)
COVID-19 related symptoms (No/Yes)	186 (84%)/36(16%)

SD: Standard Deviation, BMI: Body Mass Index, LE: Lymphoedema, BCG: Bacillus Calmette-Guérin.

## Data Availability

All data generated during this investigation are included in the main text or the Appendix A, and datasets are available upon request from the corresponding author.

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
