# Peer review of "Robust COVID-19 Vaccine Responses Despite Filarial Co-Infection: Insights from a Lymphatic Filariasis Cohort in Ghana"

_vaccines, 2025, doi:10.3390/vaccines13030312_

Round 1
Reviewer 1 Report
Comments and Suggestions for Authors
In 2000, the World Health Organization launched a global program to eliminate Lymphatic Filariasis (LF). As a result, significant progress has been made in stopping the spread of the infection through large-scale annual treatment of LF patients in regions where the infection is present. At the same time, approximately 51 million people were infected with LF pathogens in 2018. In response to the invasion of LF pathogens, as in other helminthiases, a second type of immune response (Th2, M2 macrophages, eosinophils) develops, which mutually counteracts the first type of immune response (CTL, Th1, M1). At the same time, the immune response of the first type is protective, but with high severity, it is dangerous for COVID-19. In this context, the reciprocal effects of COVID-19 (and/or COVID-19 vaccines) and LF are of great interest. In view of the above, this article may be useful to the readers of the Vaccine Journal. In general, I am positive about the results of this study, but I have a few comments:
(1) The authors note (line 180-182): "Due to limited availability and capacity of COVID-19 testing, none of the participants (n=222) had ever tested positive for SARS-CoV-2 by PCR or rapid antigen test." In this regard, the last row in Table 1 is "COVID-19 related symptoms (No / Yes)", it is unclear what the authors meant in this case: the presence of flu-like symptoms in the history or something else.
(2) In limiting their study, the authors should probably note that in their work they could not convincingly demonstrate the effect of vaccination on the incidence and severity of COVID-19 in LF, since the titers of antibodies against SARS-CoV-2 S protein are a surrogate criterion for the effectiveness of vaccines against COVID-19. The effectiveness of vaccination should be confirmed by more stringent criteria, including: the incidence of COVID-19, the severity of the course of this disease, the presence of complications and mortality. At the same time, the difference in the incidence of COVID-19 between vaccinated and unvaccinated patients is insignificant, judging by the presence of NCP-specific IgG in vaccinated patients.
(3) An even more important limitation of this study is the lack of a control group of healthy individuals, including those who received the COVID-19 vaccine and those who did not. Without this control, it is difficult to assess, for example, how much LF affects the strength of the humoral immune response to SARS-CoV-2 antigens and much more. Perhaps the authors have regional statistical data on the indicators used in the study in people who do not suffer from LF, then these data should be provided in the article.
(4) In the Conclusions section, it is probably possible to state that the use of vaccines against COVID-19 does not significantly affect the course of LF.
(5) References should be adapted to MDPI style.
Reviewer 2 Report
Comments and Suggestions for Authors
The authors conducted a study on seroprevalence of the antibodies to S- and N- antigens of SARS-CoV-2 in the Filaria-infected population from Ghana. General comment: the information provided throughout the manuscript is not logically sequential, contains repetitions and overfocusing on certain points completely omitting another aspects. Here are some specific comments for authors' consideration:
Line 57 - No COVID-19-related clinical outcomes and its relation to the parameters measured were discussed for the study population, thus please consider removing the "clinical outcomes of COVID-19" from the conclusions.
The introduction is disproportionate: it shall be more concise in the part excessively describing COVID-19 pandemic, pathogenesis, clinical pictures, vaccines etc, however more focus shall be given to often-mentioned "immunological effects of parasites". Although the closing sentence of the paper summary states about "complexity of infectious disease dynamics in co-infected populations,... filarial seropositivity influence on the immunogenicity and clinical outcomes of COVID-19", not much specific points have been described for the mechanisms of immune modulation by Filaria. The authors may consider to expand here and thus underline significance of the selected topic. Nothing about Ascaris lumbricoides is written in the introduction, although the antibodies to these species was studied and is mentioned in the key conclusions. The absence of division to paragraphs also makes the text difficult to read.
General comment to the methods section: please provide the catalog numbers and manufacturer details for the kits and reagents.
Please describe what were the criteria for classifying the patients to the unvaccinated, incomplete (I guess it means only 1 injection) and complete vaccinated group and how was the information obtained (questionnaire or documentation provided by patients or?). The vaccine used in this population also must be described in the methods section, however it only appears in the "discussion" section first.
Line 197 - please specify what is the "Ratios" and how was it calculated.
Line 206 - please provide the full name of TMB.
Lines 290, 291 - in order to justify the discrimination to the infected vs. vaccinated subjects as described, the authors should provide the data and the number/percentage of the individuals positive for anti-S antibodies but lacking anti-NC antibodies in the unvaccinated population. Such individuals are likely to be present thus questioning the approach of dividing to infected and non-infected based on anti-NC positivity only.
Lines 296-298 - What "complex interplay" are the elevated anti-N levels in the fully vaccinated group suggesting? If there are more thoughts on that, this shall be expanded and moved to discussion, if not, please consider removing this sentence.
Figure 2. Why to show the anti-spike antibody data only for the unvaccinated group and anti-NC data only for the vaccinated groups? please provide the data for all groups in the same graphical representation, currently the data shown does not support the subsection title "Elevated SARS-CoV-2 seroprevalence among unvaccinated individuals".
Lines 332-333, 455-456, 469, 482-483, 487 - "expression" is not a correct word in this context, please consider using "production" and "levels".
Figure S4 - no data is shown for the unvaccinated group, please provide it in the same representation.
Lines 345-355 - repetition of the word "potential", please consider replacing the second word to "possible".
Section 3.4 - why is the potential effect of Ascaris presence only considered for "COVID-infected" group and not for the vaccinated subjects as well? Please justify the selection. Were only the non-vaccinated population analyzed here? please describe.
Line 389 - the information that all vaccinated participants received the adenovirus-based vector vaccine should be mentioned in all the sections of the manuscript including abstract, and not here the first time.
Lines 387-393 - these lines represent a study conclusion/summary and do not belong to the results section.
Line 402, 429 - AstraZeneca is the name of the company and not of the vaccine.
Line 417-423 - again, this statement is questionable in the absence of the healthy controls or background data.
Lines 434-436 - this information is rather missing in the introduction section.
Lines 457-458 - how does this statement align with the fact that no COVID-19-related symptoms were reported in the study population?
Line 470 - this sentence is a repetition of lines 462-463.
Lines 502-503 - repetition of the lines 499-500.
Please address the issue of cross-reactivity between antibodies to Ascaris and Filaria, what magnitude does it have and what influence it may have had on the data obtained.
The word "elevated" can only be used in relation to the over time comparison or to the reference values, please revise the wording in the statements.
Conclusion section is vaguely expressed and contains very general words.
Title: the observed immune response could be rated as "robust" or "strong" only if there were a healthy control group included or if there is comparable data or the grading based on the standardized units obtained by the authors or published elsewhere. Neither was found in the present manuscript, please seek for more background information or perform additional experiments with the healthy control samples or correct the title and main conclusions such as "COVID-19 vaccination triggers a strong immune response in LF patients". One may explicitly refer to the neutralization data, however the authors do not do it and also do not discuss the limitations of the used "neutralization" assay.
Reviewer 3 Report
Comments and Suggestions for Authors
Your sample is highly skewed towards women. What is the reason for this? I did not see this discussed in the paper. Is there something about women that make the disease prevalent and does this impact your results at all?
The sample is also heavily skewed towards treatment group A. Does this impact your findings at all, and if so, how?
Your figures need to be made more clear. They were difficult to read.
What are the differences between those with flu vaccination and those not, in regards to your study? Are there any statistically significant differences?
Round 2
Reviewer 2 Report
Comments and Suggestions for Authors
The authors have addressed the comments adequately.